# SEArch: A Self-Evolving Framework for Network Architecture Optimization

## Abstract

This paper studies a fundamental network optimization problem that finds a network architecture with optimal performance (low loss) under given resource budgets (small number of parameters and/or fast inference). Unlike existing network optimization approaches such as network pruning, knowledge distillation (KD), and network architecture search (NAS), in this work we introduce a self-evolving pipeline to perform network optimization. In this framework, a simple network iteratively and adaptively modifies its structures by using the guidance from a teacher network, until it reaches the resource budget. An attention module is introduced to transfer the knowledge from the teacher network to the student network. A splitting edge scheme is designed to help the student model find an optimal macro architecture. The proposed framework combines the advantages of pruning, KD, and NAS, and hence, can efficiently generate networks with flexible structure and desirable performance. Extensive experiments on CIFAR-10, CIFAR-100, and ImageNet demonstrated that our framework achieves great performance in this network architecture optimization task.

## 1 Introduction

Deep neural networks (DNN) have achieved state-of-the-art performance in numerous computer vision and natural language processing tasks. These networks are often manually designed and fine-tuned using large datasets, resulting in complex architectures with a large number of parameters. In many applications, networks need to be simplified in order to be deployed on specific platforms with constrained resources (e.g. mobile/portable devices with limited processing/memory capability) or to meet latency requirement (e.g. for real-time tasks).

To simplify or optimize a given well-performing but oversized network, three main network optimization strategies have been widely studied in the literature: (1) network pruning (Li et al., 2016; Wang et al., 2021), (2) knowledge distillation (Gou et al., 2021; Wang & Yoon, 2021), and (3) neural architecture search (Liu et al., 2018).

- Network pruning methods iteratively and selectively remove network branches or channels according to the given budget. Although generally computationally efficient, these methods frequently lead to sub-optimal performance.
- Knowledge distillation (KD) methods transfer knowledge from a larger, trained network (the teacher) to a smaller network (the student). The presence of a teacher network enhances training efficiency. However, the architecture of the student network is usually manually predefined, without optimization, which can compromise its performance.
- Neural architecture search (NAS) methods aim to discover the most effective architecture by navigating a vast space of network components. While this approach can yield networks with adaptable structures and impressive performance, conducting a thorough search is frequently prohibitively expensive and demands substantial computational resources.

We denote the given well-trained network in need of simplification as the *teacher model* and assume that it has acquired effective feature maps. Our goal is to discover a new network, denoted as the *student model*, with a constrained number of parameters while maintaining optimal performance, such as classification accuracy. We introduce a **S**elf-**E**volving **Arch**itecture optimization framework

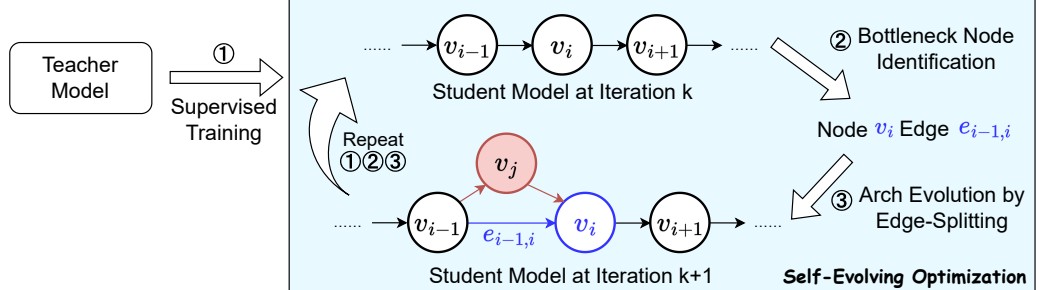

Figure 1: An overview of the proposed pipeline. Given a well-trained teacher model that requires optimization, we iteratively evolve and grow a student network until the given resource constraint is reached. In each iteration, the optimization process includes three stages: (1) transferring knowledge from the teacher model to the student model by supervised training; (2) selecting a candidate edge according to the modification value score; and (3) modifying the network structure of the candidate edge by edge splitting to enhance its expressiveness.

(SEArch) that is both effective (in terms of performance) and efficient, leveraging the strengths of these three strategies. Our **main idea** is to iteratively evolve a student network, allowing it to dynamically adjust its architecture until the given resource constraint is reached. The student network initiates with a very simple structure. During the optimization process, we iteratively identify and modify the bottlenecks under the guidance of the teacher model's feature maps.

Unlike network pruning methods that simply trim down the original network and knowledge distillation methods that refine a network with a fixed structure, our evolving approach allows for flexible adjustments to the overall network architecture. Compared with NAS methods that discover a network from a large supernet, our SEArch framework constructs an optimal architecture in a reverse manner. We iteratively identify the bottleneck of the student network and refine the corresponding structure, hence, our search demonstrates faster convergence.

Our **main contributions** are summarized as follows.

- We proposed a **S**elf-**E**volving network **Arch**itecture optimization framework, or SEArch. This framework iteratively adjusts the network architecture topology, incorporating the benefits of network pruning, KD, and architecture search methods.
- To facilitate faster transfer learning and network topology optimization, we designed an effective attention mechanism, a bottleneck identification, and an edge-splitting scheme for efficient construction of new networks.
- Comprehensive experiments demonstrated our framework achieves state-of-the-art performance when compared with existing network pruning and knowledge distillation algorithms.

## 2 RELATED WORK

**Network Pruning.** Traditional pruning approaches can be grouped into two categories: structured pruning and unstructured pruning. Structured pruning He et al. (2017; 2018a; 2019); Zhuang et al. (2020); He et al. (2020); Lin et al. (2020) removed entire layers or filters of a network to preserve its structural regularity. The result networks can be easily developed and deployed. Unstructured pruning Dong et al. (2017); Louizos et al. (2018) sparsified the convolutional weights or feature maps. The pruned networks require specific hardware to speed up the training and inference. These methods focus on removing the least significant network parameters or components. And the pruned networks often have a clear performance drop. Recent papers Dong & Yang (2019); Ye et al. (2020a;b) built a new network with flexible channels layer-by-layer and cannot optimize the global topological architectures. Hence, their expressive capability and performance are limited by this fixed architecture. DNN quantization and decomposition depend on hardware and are out of our scope.

**Knowledge Distillation.** Knowledge distillation (KD) transfers information from a trained teacher network to a (often smaller) student network to reduce network complexity. Hinton et al. (2015) introduced a knowledge distillation strategy to compress the model. The student network learns how a large network studies given tasks under this teaching procedure. Two surveys Wang & Yoon (2021); Gou et al. (2021) gave comprehensive discussions of KD methods. Recently, Lin et al. (2022)

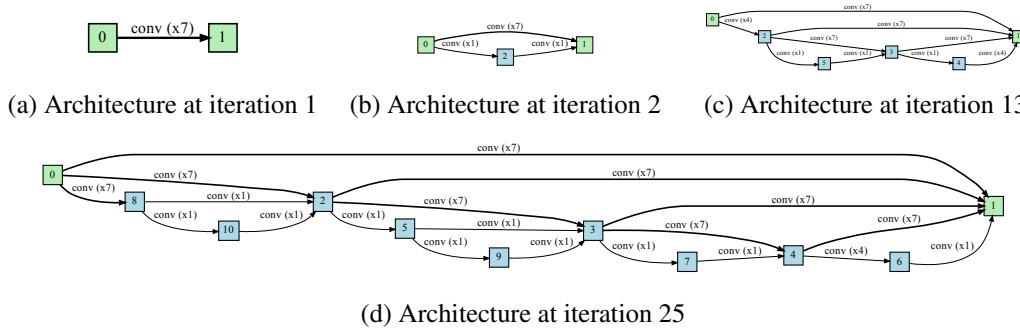

(a) Architecture at iteration 1  (b) Architecture at iteration 2  (c) Architecture at iteration 13

(d) Architecture at iteration 25

Figure 2: Visualizations of our SEArch algorithm for optimizing the architecture of student network at different iterations. The experiment was conducted on CIFAR-10. Here conv represents the $3 \times 3$ residual separable convolution. The number $(\times N)$ indicates how many stacked operations. The green nodes $v_1$ and $v_2$ are the first stage (input image) and the final stage (final feature map). (a-d) show the changes in the network architectures at different iterations.

introduced a "target-aware transformer" to align varying semantic information at the same spatial location. Dong et al. (2023) used feature semantic similarity between the teacher and student models as an indicator of distillation performance and built a training-free search framework. Inspired by existing KD methods, our approach transfers both response-based knowledge (output from the final layers) and feature-based knowledge (output from intermediate layers) to the student model. Finally, unlike existing KD methods where the student's architecture is manually designed and remains fixed during optimization, our scheme actively optimizes the student model's architecture throughout the knowledge transfer process, and this helps improve the network performance.

**Neural Architecture Search.** NAS methods typically construct a huge and comprehensive search space including many building block architectures, then search for an optimal combination of them, using reinforcement learning based (Zoph & Le, 2016; Cai et al., 2018), evolutionary algorithm based (Yang et al., 2020; Liu et al., 2020), or gradient based (Liu et al., 2018; Hu et al., 2020) algorithms. Although various approximate algorithms (e.g. weight sharing Zhang et al. (2020), proxy task Zhou et al. (2020), and cell-based Liu et al. (2018); Li et al. (2020)) have been explored to accelerate the architecture search and training, the prohibitively expensive computational cost of thorough training and searching significantly hinders the effectiveness of NAS. On the contrary, we search for the new architecture in a reverse manner that our SEArch starts from a simple network and iteratively adds new convolutional operators to improve its performance.

## 3 APPROACH

Compared with mainstream network optimization approaches that shrink existing networks or find an optimal network from a huge supernet, our self-evolving neural network optimization is in a reverse manner. It begins with a basic, primitive network (*student* model) and progressively evolves to superior ones by incorporating new convolutional operators and modifying its architectural topology. Knowledge is transferred from the teacher network to the student network through learning intermediate layers' output and final prediction. This growing scheme assists the network in better maintaining its performance, and in some cases, even surpassing that of the original network. Fig. 1 shows an overview of the proposed pipeline. In this work, we validate this design on the classic image classification task, where a color image is taken as input and its class label is predicted.

### 3.1 DEFINITIONS AND PIPELINE OVERVIEW

A neural network can be described using a directed acyclic graph $\mathcal{G} = (\mathbb{V}, \mathbb{E})$ comprising $\mathbb{V}$ nodes and $\mathbb{E}$ edges. In this context, each node $v_i \in \mathbb{V}$ represents a latent feature map, and each directed edge $e_{i,j} \in \mathbb{E}$ is associated with a specific operation $o_{i,j}$ that transforms feature map from $v_i$ to $v_j$. When a node $v_i$ has more than one incident edge, each edge generates a feature map, and the resulting feature map of $v_i$ is the sum of these individual feature maps.

Given a well-trained network, denoted as the *teacher* model, $\hat{\mathcal{G}} = (\hat{\mathbb{V}}, \hat{\mathbb{E}})$ requires optimization with respect to the number of parameters or FLOPs. We identify the longest path originating from the

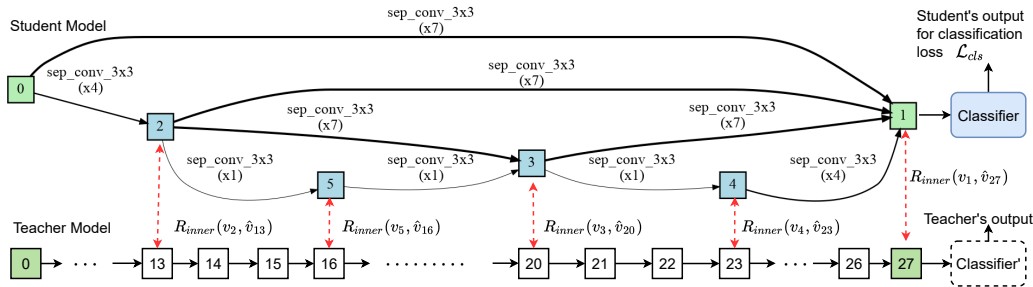

Figure 3: Illustrations of supervised learning for a student model. The student model is at the search iteration 13 on CIFAR-100. The teacher model is a well-trained ResNet-56 network that has 27 layers (at the bottom). Both models take the same input image (Node 0). The classification loss $\mathcal{L}_{cls}$ compares the outputs of the student model and the training labels. $R_{inner}$ computes the differences in feature maps between the student and the teacher, indicated by dashed red arrows.

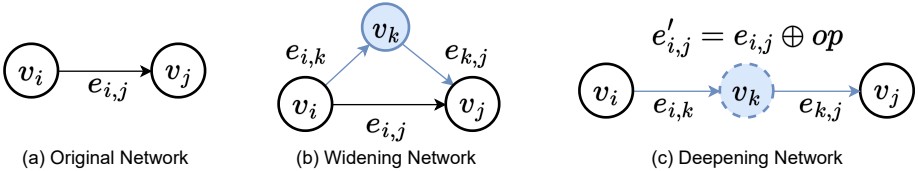

Figure 4: Illustrations of the proposed edge-splitting optimizations. (a) is the initial state of the most valuable structure for modification. (b) shows the first option by adding a supervised node $v_k$ to widen the network. (c) shows the second modification option by adding an implicit node $v_k$ to deepen the network.

input and terminating at the final feature map. The feature maps along this path constitute a list of nodes in order $\hat{\mathbb{V}} = \{\hat{v}_1, \hat{v}_2, \cdots, \hat{v}_m\}$ and the teacher model is partitioned into $m$ layers. $\hat{v}_1$ and $\hat{v}_m$ are the source and sink of the graph $\hat{\mathcal{G}}$, respectively.

The student model is initialized to a basic two-layer network, denoted as $\mathcal{G}_1 = (\mathbb{V}_1, \mathbb{E}_1)$. $\mathcal{G}_1$ comprises two nodes $\mathbb{V}_1 = \{v_1, v_2\}$, where $v_1$ represents the input image and $v_2$ is the final feature map fed into the classifier. The only edge $e_{1,2} = (v_1, v_2)$ in $\mathbb{E}_1$ denotes a convolution operation that transforms data from $v_1$ to $v_2$. Subsequently, the proposed framework iteratively optimizes the student model $\mathcal{G}_z (z \geq 1)$ until its model size reaches the budgetary limit $B$. In each iteration, the architecture of $\mathcal{G}_z$ was optimized by two stages: *training* and *self-evolving*. (1) *Training* stage transfers the knowledge from the teacher model to the student model $\mathcal{G}_z$ by supervised training (Sec. 3.2) and identifies the bottleneck to refine (Sec. 3.3). (2) *Self-evolving* stage modifies the topological structure of this bottleneck edge by edge-splitting (Sec. 3.4). We split the training dataset into two subsets for these two stages, respectively. Fig. 2 provides an example of how the student's architecture evolves during the self-evolving scheme.

## 3.2 Supervised Learning through the Teacher Model

Training student model $\mathcal{G}_z$ allows us to optimize its parameter weight and identify the bottleneck structure for architecture evolving. The student model is expected to imitate the behavior of the teacher model, including the network's output and the inner feature maps. Hence, we use the inner feature maps $\hat{\mathbb{V}}$ produced by the teacher model as checkpoints to supervise the student models' inner nodes.

In the first iteration, the student model is a two-node network with nodes $v_1$ and $v_2$. Here $v_1$ represents the input image while $v_2$ contains the final feature map. We construct a mapping table $q$ to map each student node to a teacher node in $\hat{\mathbb{V}}$. Then $q_i$ indicates that teacher node $\hat{v}_{q_i}$ transfers knowledge to the student node $v_i$ during training. For example, if $q_2 = m$, it means that the student node $v_2$ is supervised by the final layer $\hat{v}_m$ of the teacher model. When introducing a new node $v_k$ into the student model, we match it with an appropriate teacher node $\hat{v}_{k'}$, denoted by $q_k = k'$. Further details on the construction of $q$ can be found in Sec. 3.4.

The student model's weights are optimized from two perspectives using the training set. First, we minimize the classification loss $\mathcal{L}_{cls}$ that computes the differences between the student model's output and the groundtruth labels. When the task is image classification, we use Cross Entropy Loss for $\mathcal{L}_{cls}$. The second objective function aims to minimize the learning loss between the student model and the teacher model. Although student node ($v_i$) and its corresponding teacher node ($\hat{v}_{q_i}$) can have feature maps with identical height and width dimensions, they often differ in the number of channels. Inspired by Lin et al. (2022), we employ an attention module $f_a$ to facilitate the aggregation of feature channels from the teacher model for transfer learning. Unlike Lin et al. (2022), which focuses on finding matching features in spatial locations, our model aims to identify matching features in channel space. For every channel in the student's feature map, an attention query is used to calculate the corresponding channel weights in the teacher's feature map. These calculated weights are then used to project the teacher's feature maps into a feature space that matches the channel dimensions of the student model's feature map. A projected feature map for the teacher node $\hat{v}_{q_i}$ can be obtained as

$$f_a(\hat{v}_{q_i}) = \text{Atten}(v_i, \hat{v}_{q_i}, \hat{v}_{q_i}). \tag{1}$$

We use the L2 norm to measure the differences between feature maps of the student node $v_i$ and teacher node $\hat{v}_{q_i}$, denoted as

$$R_{inner}(v_i) = ||v_i - f_a(\hat{v}_{q_i})||_2^2. \tag{2}$$

Then, we define the imitation loss $\mathcal{L}_{inner}$ of the student model as the average of the difference on all student nodes,

$$\mathcal{L}_{inner} = \frac{1}{|\mathbb{V}|} \sum_{v_i \in \mathbb{V}} R_{inner}(v_i), \tag{3}$$

where $|\mathbb{V}|$ represents the number of nodes in the student model. Fig. 3 visualizes the supervised training in one iteration. We train the parameter weight of the student model by minimizing

$$\mathcal{L} = \mathcal{L}_{cls} + \mathcal{L}_{inner}. \tag{4}$$

### 3.3 BOTTLENECK IDENTIFICATION FOR ARCHITECTURE OPTIMIZATION

We model network learning as an information gain process. Here we design a scheme to identify the bottleneck of the student model and then improve its topology architecture. A bottleneck implies that making modifications to this local structure could lead to large potential performance gains. Although $R_{inner}$ indicates the difference between the student node and the teacher node, it doesn't actually reflect the bottleneck node (see the ablation study). We further define a modification value score $S$ to estimate the bottleneck node.

Considering a node $v_j$ in the student model, we define another student node $v_i$ that shares its incoming incident edge $(u_i, v_j)$ as its *precursor node*. Similarly, we define a student node $v_k$ that shares its outgoing incident edge $(v_j, u_k)$ as its *successor node*.

(1) First, if the feature map at node $v_j$ is inaccurate, it propagates this inaccuracy, affecting all its successor nodes. Hence, improving the accuracy of $v_j$ will benefit all its successor nodes. We use the out-degree $\deg_j^+$ to denote the number of successor nodes of node $v_j$. Assuming that the error $R_{inner}(v_j)$ could be minimized to a theoretical minimum of 0 after refining node $v_j$, the positive impact would be proportional to $\deg_j^+ \times R_{inner}(v_j)$.

(2) Second, to modify the feature map of node $v_j$, we often need to change the network architecture from its precursor nodes to $v_j$. Each time, we select one precursor node $v_i$ and add additional convolutional operations to increase the depths/widths of the local network, named, splitting the edge $e_{i,j}$. We use the in-degree $\deg_j^-$ to denote the number of precursor nodes of node $v_j$. Assuming each incoming edge of $v_j$ independently and evenly contributes to the error $R_{inner}(v_j)$, an improvement from modifying one of the $\deg_j^-$ incoming edges can be expected to be proportional to $\frac{1}{\deg_j^-} \times R_{inner}(v_j)$.

Finally, we define the *modification value* score $S$ for each student node $v_j$ as

$$S(v_j) = \frac{\deg_j^+}{\deg_j^-} R_{inner}(v_j), \tag{5}$$

where $R_{inner}(v_j)$ is the deviation of this node's feature map from the teacher model, and $\frac{\deg_j^+}{\deg_j}$ is the adjustment term that estimates the potential performance gain from modifying the local structure. We compute $S$ for each student node using the validation dataset. The node with the largest $S$ is selected and its local structure will be refined by edge-splitting. In our experiments, we selected the edge associated with the closest precursor node for architecture evolution.

## 3.4 Architecture Evolution by Edge Splitting

Suppose the current student network is $\mathcal{G}_z = (\mathbb{V}, \mathbb{E})$. Node $v_j \in \mathbb{V}$ and edge $e_{i,j} = (v_i, v_j) \in \mathbb{E}$ are selected for improvement by edge-splitting. Our design is based on the observation that: *Increasing the depth or width of a network can either maintain or improve output accuracy.* Hence, if the model size of a given network (e.g. number of parameters, FLOPs) has not reached the budget limit, we evolve its architecture by adding new convolutional operations to enhance both its size and performance.

To refine a local structure at node $v_j$, which has not approximated the teacher network well, we introduce a new node $v_k$ between its precursor node $v_i$ and $v_j$, and perform two types of edge-splitting: widening the network and deepening the network.

(1) *Widening the network.* One option is to create a new branch beside $e_{i,j}$ by inserting a new node $v_k$, as illustrated in Fig. 4(b). Two new convolutional operations are also added as $e_{i,k}$ and $e_{k,j}$. This operation increases the width of the student network to improve its capability. We select the one of intermediate layers between $q_i$ and $q_j$ as the guidance for $v_k$ :

$$q_k = \lfloor c \cdot q_i + (1 - c) \cdot q_j \rfloor, \tag{6}$$

where $c \in [0, 1]$ is a hyper-parameter. In our experiment, we have tested that $c = 0.5$ works the best. This can be explained by the intuition that the information gain of a shallow network is proportional to its depth. Thus the total information gain can be represented as $c(1 - c)$, which is maximized at $c = 0.5$. We also discussed the comparisons of choosing $c$ in the ablation study.

(2) *Deepening the network.* The second option is to deepen the network by adding a new node $v_k$ between $v_i$ and $v_j$ with new convolutional operations. To reduce the network latency, we didn't assign a teacher node to supervise the training of node $v_j$. In other words, this operation can be considered as stacking/concatenating a new convolutional operation $op$ to the old ones on $e_{i,j}$, as illustrated in Fig. 4(c).

While either widening or deepening the local structure could increase the capability of the network, deepening operation is simpler and introduces less computation overhead. In our self-evolving framework, we first use the deepening modification until the number of stacked operations reaches a predefined number $B_{op}$. Algorithm 1 summarizes the proposed self-evolving framework.

## 3.5 Convolutional Operation

We perform the network optimization through self-evolving by incorporating the network creation capability from neural architecture search. Most architecture search algorithms first define a set of candidate operations $O$, then search and pick the best operations during optimization. Having a big operation set provides more choices and bigger diversity in designing networks, but it significantly slows down the search runtime. Recent study Yang et al. (2019) suggested that when searching a neural architecture, the quality of macro-structures (edge connections) is more important than micro-structures (operations). In this work, we only use $3 \times 3$ separable convolution as $op$ to build the network. The expressivity analysis can be found in the Appendix.

---

**Algorithm 1** Self-Evolving Arch Overview

**Input:** Teacher model $\hat{\mathcal{G}}$, training dataset $D_T$, valid dataset $D_V$, budget $B$, $B_{op}$, and Conv $op$
**Output:** Student model $\mathcal{G}$
1: $\mathcal{G}_1 \leftarrow$ two-layer network, $z \leftarrow 1$
2: **while** $|\mathcal{G}_z| < B$ **do**
3:      Train the student model $\mathcal{G}_z$ on $D_T$ by Eq. 4
4:      Select node $v_j$ and $e_{i,j}$ on $D_V$ by Eq. 5
5:      **if** $|e_{i,j}| < B_{op}$ **then**
6:          Deepening. $e_{ij}' = e_{ij} \oplus op$
7:      **else**
8:          Widening. Add node $v_k, e_{i,k}, e_{k,j}$
9:          Set supervised $q_k$ from $\hat{\mathcal{G}}$ by Eq. 6
10:      **end if**
11:      $z = z + 1$
12: **end while**

---

Table 1: Ablation study on CIFAR-10. "Base Acc." and "Pruned Acc" are accuracy of the baseline and optimized networks. "Acc. Drop" is the accuracy drop (smaller is better), where a negative value means the optimized model outperforms the baseline. "PARAMs" is the network parameters.

| Method | Base Acc. (%) | Pruned Acc. (%) | Acc. Drop (%) | PARAMs (M) | FLOPs (M) |
|---|---|---|---|---|---|
| Baseline: ResNet-56 | 93.95 | - | - | 0.85 | 126.6 |
| (A) Ours w/o $S$ (Eqn. 5) | 93.95 | 84.18 ($\pm$3.34) | 9.77 | 0.41 | 90.9 |
| (B) Ours $c = 0.25$ (Eqn. 6 | 93.95 | 94.62 | -0.67 | 0.40 | 78.9 |
| (B) Ours $c = 0.75$ (Eqn. 6 | 93.95 | 94.50 | -0.55 | 0.40 | 142.1 |
| (C) Random Splitting (0.40) | 93.95 | 91.79 ($\pm$2.14) | 2.16 | 0.40 | 63.1 |
| **Our full model (0.40)** | 93.95 | **94.82** ($\pm$0.14) | **-0.87** | 0.40 | 63.1 |
| Baseline: ResNet-110 | 94.04 | - | - | 1.73 | 255.0 |
| (C) Random splitting (0.68) | 94.04 | 92.81 ($\pm$0.52) | 1.22 | 0.68 | 85.5 |
| **Our full model (0.68)** | 94.04 | **95.00** ($\pm$0.02) | **-0.97** | 0.68 | 85.5 |

Table 2: Comparisons with network pruning methods for optimizing ResNet-56 on CIFAR-10. The "FLOPs $\downarrow$" is the pruned ratio on FLOPs. The fields have the same meaning as Table 1.

| Method | Base Acc. (%) | Pruned Acc. (%) | Acc. Drop (%) | FLOPs (M) | FLOPs $\downarrow$ (%) |
|---|---|---|---|---|---|
| MIL (Dong et al., 2017) | 94.35 | 92.81 | 1.54 | 78.1 | 37.9 |
| Polar (Zhuang et al., 2020) | 93.80 | 93.83 | -0.03 | - | 47.0 |
| AMC (He et al., 2018b) | 92.80 | 91.90 | 0.90 | 62.7 | 50.0 |
| HRank (Lin et al., 2020) | 93.26 | 93.17 | 0.09 | 62.7 | 50.0 |
| Greg-1 (Wang et al., 2021) | 93.36 | 93.06 ($\pm$0.09) | 0.30 | 62.0 | 50.2 |
| SFP (He et al., 2018a) | 93.59 | 93.35 ($\pm$0.31) | 0.24 | 59.4 | 52.6 |
| FPGM (He et al., 2019) | 93.59 | 93.26 ($\pm$0.03) | 0.33 | 59.4 | 52.6 |
| TAS (Dong & Yang, 2019) | 94.46 | 93.69 | 0.77 | 59.5 | 52.7 |
| LFPC (He et al., 2020) | 93.59 | 93.24 ($\pm$0.17) | 0.35 | 59.1 | 52.9 |
| ResRep (Ding et al., 2021) | 93.71 | 93.71 ($\pm$0.02) | 0.00 | 59.1 | 52.9 |
| **Ours (0.40)** | 93.95 | **94.82** ($\pm$0.14) | **-0.87** | 63.1 | 50.2 |

## 4 EXPERIMENTS

We evaluated our SEArch algorithm using image classification task and compared it with state-of-the-art network optimization approaches: network pruning and knowledge distillation. Due to the page limit, we discuss the main results here and move the implementation details, time complexity and runtime analysis to the Appendix.

### 4.1 ABLATION STUDY

We conducted ablation studies on the CIFAR-10 dataset to evaluate the effectiveness of each component in the proposed SEArch model.

**Effectiveness of Modification Value Score in Eqn. 5.** The results are reported in Table 1 Exp (A). We removed the adjustment term $\frac{\deg^+}{\deg^-}$ in Eqn. 5, and the rest parts are the same as our full model. With this setting, the bottleneck node was selected with the largest $R_{inner}$. The accuracy of the optimized network drops $0.8\%$, while our full model has $0.87\%$ accuracy gain. It shows that our modification value score in Eqn. 5 is critical to identify the bottleneck to guide the architecture modification.

**Different Supervision Layer of Teacher Model in Eqn. 6.** In Eqn. 6, the parameter $c$ determines the index of intermediate layers of the teacher model that is assigned to supervise the new node in the student model. We conducted experiments by setting $c = 0.25$ (close to the precursor node) and $c = 0.75$ (close to the bottleneck node). In Table 1 Exp (B), the results of $c = 0.25$ and $c = 0.75$ are slightly weaker than our full model ($c = 0.50$). We concluded that our SEArch is robust to different

Table 3: Comparisons with pruning methods for optimizing ResNet-56 on CIFAR-100. The fields have the same meaning with Table 2.

| Method | Base Acc. (%) | Pruned Acc. (%) | Acc. Drop (%) | FLOPs (M) | FLOPs ↓ (%) |
|---|---|---|---|---|---|
| Polar (Zhuang et al., 2020) | 72.49 | 72.46 | 0.06 | - | 25.0 |
| OICSR-GL (Li et al., 2019) | 75.87 | 76.23 | -0.66 | - | 38.5 |
| MIL (Dong et al., 2017) | 71.33 | 68.37 | 2.96 | 76.3 | 39.3 |
| TAS (Dong & Yang, 2019) | 73.18 | 72.25 | 0.93 | 61.2 | 51.3 |
| LFPC (He et al., 2020) | 71.41 | 70.83 | 0.58 | 60.8 | 51.6 |
| SFP (He et al., 2018a) | 71.40 | 68.79 | 2.61 | 59.4 | 52.6 |
| FPGM (He et al., 2019) | 71.41 | 69.66 | 1.75 | 59.4 | 52.6 |
| **Ours (0.80)** | 70.79 | 73.86 (±0.14) | **-3.08** | 86.8 | 31.8 |
| **Ours (0.40)** | 70.79 | 73.00 (±0.20) | **-2.20** | 56.9 | 55.3 |

Table 4: Comparisons with pruning methods for optimizing ResNet-50 on ImageNet

| Method | Top-1 Acc. (%) | Top-5 Acc. (%) | PARAMs (M) |
|---|---|---|---|
| ResNet50 | 76.15 | 92.87 | 25.56 |
| HRank (Lin et al., 2020) | 71.98 | 91.01 | 13.77 |
| ThiNet-50 (Luo et al., 2017) | 71.01 | 90.02 | 12.38 |
| GAL-1-joint (Lin et al., 2019) | 69.31 | 89.12 | 10.21 |
| GDP-0.5 (Lin et al., 2018) | 69.58 | 90.14 | - |
| Taylor-FO (Molchanov et al., 2019) | 71.69 | - | 7.90 |
| S-ResNet-50 (Yu et al., 2018) | 72.10 | 90.57 | 6.92 |
| **SEArch (Ours)** | 72.55 | 90.81 | **4.98** |

settings of $c$. Besides the information gain process intuition, the experiment supports choosing the middle layer of the teacher model provides the best results.

**Bottleneck Identification v.s. Random Edge-splitting.** A random edge-splitting setting is to randomly pick edges from the student model, and randomly add new operations during iterative optimization until the model reaches the predefined model size budget. We ran the random edge-splitting three times and recorded the average results. With the proposed bottleneck identification strategy, our SEArch model is able to select key nodes for architecture optimization. Table 1 Exp (C) reports the performance of the random splitting strategy and our full SEArch model. When the baseline is ResNet-56, the random splitting strategy has a 2.16% accuracy drop while our model achieved 0.87% accuracy improvement from the baseline model. When the baseline is ResNet-110, the random splitting strategy has 1.22% accuracy drop while our model yields 0.97% accuracy gain.

## 4.2 COMPARISONS WITH NETWORK PRUNING METHODS

**Results on CIFAR-10.** We follow the common evaluation settings of network pruning papers. On CIFAR-10, we evaluated our SEArch algorithm on ResNet with depths 56 and 110. Note that the weights of the baseline models vary in these papers. We chose a baseline model with relatively high accuracy for fair comparisons because when started with a more accurate model, it is harder for pruning to maintain the accuracy. We chose the widely adopted pretrained model Chenyaofo (2021) as the baseline model. We set the number of parameters as our pruning target as 0.40M. Following

Table 5: Comparisons with KD methods on CIFAR-10

| Method | Acc. (%) |
|---|---|
| KD (Hinton et al., 2015) | 92.63 |
| AT (Zagoruyko & Komodakis, 2016) | 92.87 |
| FT (Kim et al., 2018) | 93.15 |
| OD (Heo et al., 2019) | 93.19 |
| Tf-KD(S) (Yuan et al., 2020) | 92.59 |
| CRD (Tian et al., 2019) | 93.20 |
| IE-AT (Huang et al., 2021) | 93.30 |
| IE-FT (Huang et al., 2021) | 93.43 |
| IE-OD (Huang et al., 2021) | 93.47 |
| **Ours (0.27M PARAMs)** | **93.58** |

experiments done in other pruning papers, we ran the model three times and reported the "mean ($\pm$ std)" results. The comparisons of optimizing ResNet-110 are discussed in the Appendix.

The quantitative comparisons of optimizing ResNet-56 are shown in Table 2. The proposed SEArch outperforms the existing pruning methods. Our method pruned the ResNet-56 by 50.2% FLOPs, and the optimized networks actually surpassed the baseline by 0.87% in accuracy. Conventional pruning methods remove redundant filters and trim the network, and the accuracy of the pruned network is often worse than the original network. The reason is that they use linear representation to approximate the original network, which could lose accuracy during feature reduction.

**Results on CIFAR-100.** The quantitative comparisons of pruning ResNet-56 on CIFAR-100 are shown in Table 3. Note that the weights of the baseline models vary in these papers, we chose our baseline from a public repository for fair comparisons. Our method pruned the ResNet-56 by 31.8% and 55.3% FLOPs, the optimized networks improved better accuracy of 3.08% and 2.20% over the baseline model. While existing methods such as SFP He et al. (2019) and Polar Zhuang et al. (2020) have 1.75% and 0.06% accuracy dropped from the baseline model. Fig. 2 visualizes how the proposed SEArch framework optimizes the architecture of the student model (0.40M).

**Results on ImageNet.** We also conducted experiments for ResNet-50 optimization on ImageNet. The results are shown in Table 4. We follow the mobile setting and set our maximum parameter count to 5.0M. Compared with the recent optimization methods, our method optimized the baseline network to a extremely small size (from 25.56M to 5.0M, 20% size of the original network), while preserved comparable Top-1 and Top-5 accuracy.

### 4.3 COMPARISONS WITH KNOWLEDGE DISTILLATION METHODS

Following the common settings in KD papers, we conducted experiments on CIFAR-10 dataset. We selected ResNet-56 as the teacher model and transferred the knowledge to a smaller network. Quantitative comparisons are reported in Table 5. State-of-the-art KD papers selected ResNet-20 as the student model and the architecture optimization is not considered during knowledge transfer. In contrast, our method integrates the architecture optimization technique to search for the optimal architecture for the student network. Because ResNet-20 has 0.27M parameters, we set the target parameters of the searched network to 0.27M for fair comparisons. Under the same parameter count, the optimized architecture by our SEArch has the best accuracy 93.58%, which outperforms the existing KD papers. Both KD methods and ours require training on the dataset to transfer the knowledge, our method requires longer time to optimize the architecture (see the Appendix for runtime analysis) but achieves better accuracy.

## 5 CONCLUSIONS

We present a self-evolving neural network optimization framework that combines the strengths of network optimization methods (i.e. network pruning, KD, and NAS). Starting from a basic structure, our student model iteratively identifies bottlenecks in the network and refines its architecture under the guidance of the teacher network. Our adoption of a single operation and the design of edge splitting enable more efficient local structure creation and topology modification. Experiments on image classification demonstrated the effectiveness of our approach.

**Limitations and Future Work.** The proposed framework optimizes an existing network in a manner similar to network pruning and KD. When compared to NAS, our approach achieves quicker optimization, but sometimes lower accuracy. Using a teacher model for guidance to identify bottlenecks in the evolving network architecture reduces the search cost but may result in sub-optimal network structures. Although the student model is capable of self-evolving to improve its architecture, its performance might be constrained by the teacher model. To eliminate the need for such a prior, we plan to explore new metrics based on information gain theory for better guidance in autonomous network evolution.

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

## A    SINGLE CONVOLUTION EXPRESSIVITY ANALYSIS

We observed that the operations of DARTS Liu et al. (2018) can be replaced using a single convolutional operation such as the $3 \times 3$ separable convolution. First, $3 \times 3$ average pooling, identity (skip connection), and $zero$ are in the same group, which are special cases of the $3 \times 3$ convolution with specific weights. Second, following ResNet, $3 \times 3$ max pooling can be replaced by a $3 \times 3$ convolution with $stride = 2$. Third, $5 \times 5$ or $7 \times 7$ convolutions can be approximated by stacked $3 \times 3$ convolutions. Last, dilated separable convolutions are special cases of separable convolutions with larger kernel sizes and zero parameter weight. As results, $3 \times 3$ and $5 \times 5$ separable convolutions, $3 \times 3$ and $5 \times 5$ dilated separable convolutions can be induced to stacked $3 \times 3$ separable convolutions. Hence, using just a single operation greatly reduces the computational cost of operation search, yet still offers similar model expressivity.

## B    IMPLEMENTATION DETAILS

We implemented our framework on PyTorch. The convolutional operation we choose is the $3 \times 3$ residual separable convolution, or sep_conv_3x3 for short. Fig. 5 shows the conventional convolutional unit stacking two full convolutions (named Conv 3x3). ResNet He et al. (2016) improves its performance by adding a skip connection (indicated by the blue line). MobileNet Sandler et al. (2018) found that depth-wise separable convolution is more efficient than full convolutions. We follow their design and replace the full convolutions with two layers: (1) The first layer is a depth-wise convolution that performs lightweight filtering (named DW Conv 3x3). (2) The second layer is a point-wise convolution that computes a linear combination of the input channels (named Conv 1x1). Fig. 6 illustrates the detailed structure of the $3 \times 3$ residual separable convolution.

The max stacked operations per edge $B_{op}$ is set to 7. To speed up the search, on each edge, we initialize the number of operations as 1, then we append 3 operations once when an unsupervised node is added. The training epoch number for each training stage is 10, which is usually good enough to reveal the underperforming nodes in the current student model. We stop the self-evolving procedure when the size of the student model reaches the limit. Following the training settings of existing papers on CIFAR-10, the final student model is retrained for 400 epochs. We choose SGD with a momentum of 0.9 and a weight decay of 0.0003 in all the training. The initial learning rate is set to 0.025 and the step scheduler is used to tune the learning rate during training. On ImageNet, we chose learning rate 0.01 and trained the model for 100 epochs.

## C    TIME COMPLEXITY ANALYSIS

Given an architecture with $V$ nodes and $E$ directed edges, if we have $K$ candidate operations, the time complexity of the operation search is $O(T_{op}) = O(K^{|E|})$. Even a relatively small $K$ makes this search expensive, and it limits the size of edges $E$ a generated network can have. To accommodate the huge operation search cost, existing NAS methods often restrict the search and structural optimization in cell or block levels. Then, the design of connections between cells/blocks is done manually based on expert knowledge and enormous experiments.

In contrast, as we observed, this operation search is unnecessary. We use a single operation to build the architecture and can then perform a more comprehensive search on the network's global structure. We use $T_{arch}$ to represent the time of architecture search and $T_w$ to represent the time of training parameter weights. With the single candidate operation selected, $K = 1$, our method reduces the overall time complexity from $O(T_{arch}T_{op}T_w) = O(T_{arch}K^{|E|}T_w)$ to $O(T_{arch}T_w)$, which is significantly smaller with even a moderate size of $|E|$.

## D    RUNTIME ANALYSIS

Experiments from other papers were performed on different hardware configurations, we found it very difficult to directly compare the runtime performance. Since the proposed SEArch framework optimizes architecture starting from a primitive one, when the target model size is much smaller than the baseline model size, our algorithm needs less time to prune the network. For example,

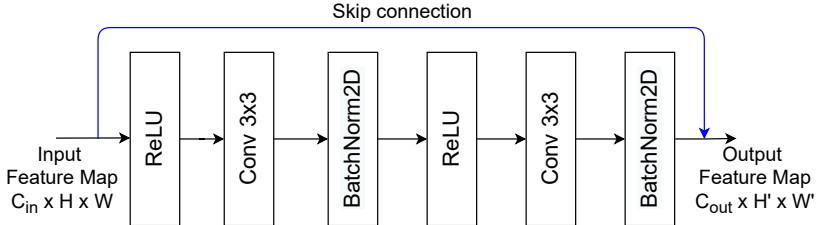

Figure 5: Conventional convolution unit stacking two full convolutions. ResNet improves its performance by adding a skip connection (blue line).

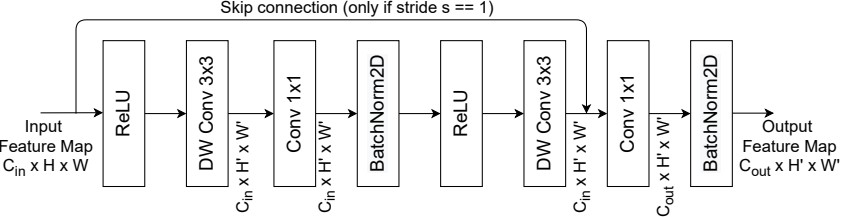

Figure 6: Convolution unit of the residual separable convolution $3 \times 3$, named sep_conv_3x3. DW Conv is the standard depth-wise separable convolution. $C_{in}$ and $C_{out}$ represent the feature channels of the input and output feature maps. The stride $s$ equals either 1 or 2, which controls spatial size of the output feature map, $H' = H/s$ and $W' = W/s$.

recent network pruning literature TAS Dong & Yang (2019) reported their runtime statistics. For the CIFAR-10 dataset, TAS searched a network with 92.65% accuracy, and it cost 10.6 GPU hours on an NVIDIA V100 GPU. In contrast, our method SEArch searched a network with **94.01% accuracy**, and it cost only **4.0 GPU hours** on an NVIDIA 1080Ti GPU, which reduced 62.3% search time. Additionally, the training speed of V100 is $100\%$ faster than 1080Ti in general. Hence, **our proposed scheme can more efficiently optimize neural architectures**.

Comprehensive runtime analysis of our SEArch is reported in Table 6. We ran our pipeline on a single NVIDIA 1080Ti GPU (11GB MEM). The running time depends on the baseline model size and target model size. Larger models require longer search/retraining time. Our model has small runtime and memory footprints. Note that **our SEArch framework is memory-friendly**: it can fit into a 4GB-MEM GPU when the batch size is set to 32.

## E    COMPARISONS WITH NETWORK PRUNING ON RESNET-110 ON CIFAR-10

Recent network pruning methods also conducted experiments on optimizing ResNet-110 on CIFAR-10. The qualitative comparison is shown in Table 7. Note that in this dataset, reducing the PARAMs to a half size is a common setting in pruning literature, and this corresponds to 0.86M PARAMs. In addition, we also cut $60\%$ off of FLOPs, which resolves to 0.68M PARAMs, for a fair comparison with HRank (Lin et al., 2020). Our method optimized the ResNet-110 by 61.2% and 66.4% FLOPs, the optimized networks improved the accuracy by 1.15% and 0.97% over the baseline model. Fig. 7 plots the accuracy gain of the pruned network over the pruned ratio on FLOPs. Our method outperforms existing network pruning methods by achieving bigger accuracy gains and a higher pruned ratio on FLOPs.

| Dataset | CIFAR-100 | | CIFAR-10 | | | |
|---|---|---|---|---|---|---|
| Baseline Model | ResNet-56 | | ResNet-56 | | ResNet-110 | |
| PARAMs (M) | 0.40 | 0.80 | 0.40 | 0.80 | 0.68 | 0.86 |
| Search Time | 3.7 / 24 | 12.7 / 38 | 4.0 / 25 | 18.3 / 47 | 11.8 / 38 | 13.5 / 44 |
| Retraining Time | 8.7 / 400 | 19.0 / 400 | 10.1 / 400 | 22.4 / 400 | 15.6 / 400 | 17.2 / 400 |

Table 6: Runtime results under different pruning settings. The codes are implemented on PyTorch. The experiments are conducted on Ubuntu 18.04 with a single NVIDIA 1080Ti GPU. Our SEO model can optimize a network at low computation costs. (GPU Hours / Iters)

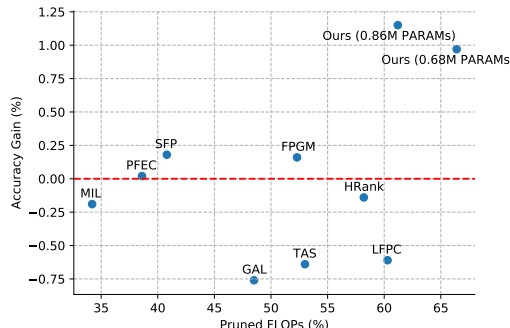

Figure 7: The accuracy gain of the optimized network over the optimized ratio on FLOPs. We compare the optimized networks from ResNet-110 on CIFAR-10. The accuracy gain means the accuracy improvement after pruning. Detailed comparisons are reported in Table 7. The red horizontal line indicates the zero accuracy gain. Our optimized networks are on the right top positions that outperform the state-of-the-art network pruning methods.

Table 7: Quantitative comparisons with network pruning methods for optimizing ResNet-110 on CIFAR-10. "Base Acc." and "Pruned Acc" are accuracy of the baseline and optimized networks. "Acc. ↓" is the accuracy drop (smaller is better), where a negative value means the result model outperforms the baseline. The "PARAMs ↓" is the pruned ratio on network parameters. The "FLOPs ↓" is the pruned ratio on FLOPs. The Baseline of NetSlim is ResNet-164.

| Method | Base Acc. (%) | Pruned Acc. (%) | Acc. ↓ (%) | PARAMs (M) | PARAMs ↓ (%) | FLOPs (M) | FLOPs ↓ (%) |
|---|---|---|---|---|---|---|---|
| NetSlim | 94.58 | 94.73 | -0.15 | 1.10 | 35.2 | 275 | 44.9 |
| MIL | 93.63 | 93.44 | 0.19 | - | - | 166 | 34.2 |
| PFEC | 93.53 | 93.30 | -0.02 | 1.16 | 32.4 | 155 | 38.6 |
| SFP | 93.68 | 93.86 (±0.21) | -0.18 | - | - | 150 | 40.8 |
| GAL | 93.5 | 92.74 | 0.76 | 0.95 | 44.8 | 130.2 | 48.5 |
| FPGM | 93.68 | 93.74 (±0.10) | -0.16 | - | - | 121 | 52.3 |
| TAS | 94.97 | 94.33 | 0.64 | - | - | 119 | 53.0 |
| HRank | 93.50 | 93.36 | 0.14 | 0.70 | 59.2 | 105.7 | 58.2 |
| LFPC | 93.68 | 93.07 (±0.15) | 0.61 | - | - | 101 | 60.3 |
| ResRep | 94.64 | 94.62 (±0.02) | 0.02 | - | - | - | 58.2 |
| LCAF | 93.77 | 93.92 | -0.15 | - | 60.1 | - | 59.8 |
| **Ours (0.86)** | 94.04 | 95.19 (±0.05) | **-1.15** | 0.86 | 50.2 | 98.8 | 61.2 |
| **Ours (0.68)** | 94.04 | 95.00 (±0.02) | **-0.97** | 0.68 | 60.4 | 85.5 | 66.4 |

