# OpenReview forum: "SEArch: A Self-Evolving Framework for Network Architecture Optimization"
_ICLR.cc/2024/Conference — Submitted to ICLR 2024_

### Official Review · Reviewer_6dDt · 2023-11-01

**Soundness:** 2 fair
**Presentation:** 3 good
**Contribution:** 2 fair
**Rating:** 5
**Confidence:** 4

**Summary:**

This paper proposes a self-evolving framework called SEArch to optimize neural network architectures in an efficient manner. The key ideas are to start from a simple network and iteratively identify bottlenecks and refine the structure under guidance from a teacher network through knowledge distillation. The approach leverages the strengths of network pruning, knowledge distillation, and neural architecture search.

**Strengths:**

1. The proposed SEArch framework combines insights from multiple areas in a novel way.

2. The approach of starting from a simple network and iteratively identifying bottlenecks to evolve the architecture is an interesting idea.

3. The experiments demonstrate that SEArch achieves better performance compared to baseline methods.

**Weaknesses:**

1. The single convolutional operation used may limit the expressivity of the generated architectures. In these days, more advanced models are dominated by self-attention and transformers models. It would be much better to use some transformer models for validation, like ViT.

2. The guidance from the teacher network could potentially constrain the optimized student network's performance, as stated by the limitation by the authors. A kind suggestion is that the teacher models could be ViT Large to be a stronger guidance.

**Questions:**

Please see the weakness.

---

### Official Review · Reviewer_rdLr · 2023-11-02

**Soundness:** 3 good
**Presentation:** 3 good
**Contribution:** 1 poor
**Rating:** 3
**Confidence:** 3

**Summary:**

This paper proposed SEArch for neural architecture optimization. Based on a teacher model's supervision and a base architecture, SEArch gradually optimizes the architecture through 2 operations: 1) widening and deepening. The paper claims the proposed method is computationally efficient and can achieve SOTA performance.

**Strengths:**

1. The proposed method, SEArch, is new to me. The bottleneck identification part is interesting and different from the existing pruning method.
2. The presentation is clear and easy to follow.

**Weaknesses:**

1. The experiments are not convincing. The paper has compared SEArch with the pruning methods. However, SEArch used extra KD supervision while other pruning methods may not use such supervision. Additionally, how does the computational overhead of SEArch compare to pruning methods or NAS approaches? I think a thorough comparison of NAS, pruning, and KD methods in terms of accuracy and computation is needed.
2. The application of the proposed method seems to be quite limited.  1) The architecture choice is limited. The paper only uses ResNet for experiments while more advanced mobile architectures are not considered such as MobileNet or Transformers. 2) The operation set is limited. We can only widen and deepen the base architecture.

**Questions:**

1. What Eq1 exactly means? You don't define the Atten operation in the paper and I can't find the definition from the referred paper (Lin et al. (2022),).
2. In your experiments, you use the ResNet architecture as the base network. As the proposed method can only widen and deepen the network (increase the parameters/flops), how do you reduce the parameters/flops of the base network?

---

### Official Review · Reviewer_xUVA · 2023-11-03

**Soundness:** 3 good
**Presentation:** 2 fair
**Contribution:** 2 fair
**Rating:** 5
**Confidence:** 4

**Summary:**

This paper proposes a self-evolving framework to perform network optimization, which iteratively refines the architecture by identifying and addressing network bottlenecks under the guidance of a teacher network. The framework adjusts the structure by widening or deepening the network at the bottleneck to improve network capacity. This iterative process continues until the resource budget is exhausted. The framework is evaluated on CIFAR-10, CIFAR-100, and ImageNet datasets and achieves better performance compared to other KD and network pruning methods. Abalation studies are also conducted to show the effectiveness of bottleneck identification and edge splitting in the framework.

**Strengths:**

1. The bottleneck identification and edge splitting techniques are interesting and intuitive.
2. The proposed framework combines techniques from KD and network pruning, offering a fresh angle to NAS.

**Weaknesses:**

1. Most of the experiments are conducted on small datasets (CIFAR-10, CIFAR-100) with only a single experiment performed on the larger ImageNet dataset. The authors may conduct more experiments on large and diverse datasets.
2. The comparison with KD and pruning methods is limited to slightly outdated works, with the most recent being from 2021 (Table 2, 5) and 2020 (Table 3, 4). More recent works should be included for comparison.
3. The method for determining the channels and strides for each layer, and for matching the size of feature maps to the teacher network, is not explicitly described. The attention module can be used to match channels, but not height and width.
4. Some important experimental settings are missing. For instance, the authors do not mention whether data augmentation was used for retraining on CIFAR. Additionally, the experimental settings for ImageNet appear overly simplified.
5. The authors measure network efficiency based on the number of parameters and FLOPs. However, these two metrics may not accurately represent the network's actual latency or throughput. The extensive use of separable convolutions might yield a small number of parameters and FLOPs, but it could potentially increase latency, especially when compared to normal convolutions.

**Questions:**

See the weakness above

---

### Meta-Review · Area_Chair_Nrbk · 2023-12-10

**Metareview:**

The submission proposes a self-evolving framework for network optimization, in which a small neural network iteratively modifies it structure using the guidance from a teacher network. This works by identifying bottleneck locations which hinder learning and loss reduction and then adding extra capacity in those regions.
The initial ratings were 2 x 5 and 1 x 3. The reviewers raised a number of questions and issues, and no rebuttal was submitted. As a result, the final ratings are negative-leaning. The AC did not find sufficient reason to overturn this consensus.

**Justification For Why Not Higher Score:**

Initial ratings were 2 x 5 and 1 x 3. No rebuttal was submitted.

**Justification For Why Not Lower Score:**

N/A

---

### Decision · Program_Chairs · 2024-01-16

Reject